# Comparative Study of the Potentially Toxic Elements and Essential Microelements in Honey Depending on the Geographic Origin

**DOI:** 10.3390/molecules27175474

**Published:** 2022-08-26

**Authors:** Magdalena Ligor, Tomasz Kowalkowski, Bogusław Buszewski

**Affiliations:** 1Department of Environmental Chemistry and Bioanalytics, Faculty of Chemistry, Nicolaus Copernicus University, 7 Gagarina Str., 87-100 Torun, Poland; 2Interdisciplinary Centre of Modern Technologies, Nicolaus Copernicus University, 4 Wileńska Str., 87-100 Torun, Poland

**Keywords:** honey, potentially toxic elements, ICP-MS

## Abstract

The profiling and quantification of potentially toxic elements (PTEs) in honey from Poland was the main aim of this work. Due to the differences in botanical and geographical origin, 33 honey samples from various parts of Poland have been tested and compared to 12 samples taken from other countries, such as Australia, Bulgaria, Italy, Germany, Portugal, Romania and Turkey. The studied elements in honey samples were: As, Be, Cd, Co, Cr, Cu, Fe, Mg, Mn, Mo, Ni, Pb, Sb, V and Zn. In most cases, the analyzed samples of honey were characterized by the moderate values of analyzed PTEs. Only a few samples contained higher concentrations of copper and manganese were noted. The presence of cadmium and lead in the level below the background equivalent concentrations was measured in the tested samples.

## 1. Introduction

The chemical composition of honey confirms that it is a very complex matrix. It can be evidenced by over 300 components determined in different types of honey belonging to various chemical groups of compounds. The composition of honey depends on many factors, such as climatic (insolation, humidity) and environmental conditions, available plants, a botanical and geographical origin, welfare of bees and many others. It is difficult to find honey characterized by the same qualitative and quantitative compositions, even within the same variety [1,2,3,4,5]. Nevertheless, honey is an infused solution of sugars in water. In addition, it also includes other valuable contaminants, such as proteins, vitamins, minerals, enzymes (invertase, lactase, α- and β-amylase, glucose oxidase, catalase and phosphatase), flavoring compounds, free amino acids, organic acids (lactic, malic, malic, formic, citric, acetic, butyric, p-amino-benzoic, pyroglutamic and gluconic) and volatiles organic compounds [5,6,7,8,9]. According to the data available in references [1,2,3,4,5], the approximate percentage composition of honey related to its physicochemical properties has been presented in Figure 1.

In the composition of honey, some biologically active compounds, which could be treated as compounds with antioxidant, anti-inflammatory, antimicrobial, antiviral and even healing properties phytochemicals, such as polyphenols especially flavonoids, have been found [3]. Moreover, secondary undesirable components or pollutants are accumulated in honey, especially due to the long-term storage or heating. Among other undesirable compounds, 5-hydroxymethylfurfural (HMF), which is formed through the Maillard reaction from reducing sugars in acidic environments, can be mentioned [10].

Moreover, honey contains several minerals recognized as macroelements, microelements and trace elements, as well as heavy metals related mainly to environment pollution. In general, the variety of elements in honey samples largely depends on the composition of flowers, with regard to their botanical and geographical origin, as well as on the contamination degree of the natural environment in which bees live [9,11,12,13,14,15,16,17]. Honey is a product of various chemical composition, but it depends on the types and species of plants and the degree of environmental pollution in the area where nectar collection takes place. A few metals may come from external sources, such as industrial smelter pollution, industrial unit emissions, and improper procedures during honey processing and maintenance stages. Honey contains all the basic minerals obtained from nectar and honeydew, which is a sweet, sticky substance excreted by aphids and often deposited on leaves and stems. The honeydew is also secreted by some scale insects as they feed on plant sap. It is classified as a sugar-rich sticky liquid [18]. Generally, the total concentration of minerals in honey ranges from 0.1% to even above 1% [19,20]. Mineral salts present in honey affect their nutritional and health value and may significantly contribute to supplementing the deficiencies of certain elements in the human diet, especially iron, magnesium and manganese [9]. Heavy metals and elements, such as cadmium, copper, lead, zinc and arsenic, can also accumulate in honey, which is related to the contamination of the area where bees collect nectar and where honey plants grow [9]. The heavy metal content of honey can be used as an indicator of the environment pollution state. Bees transport various pollutants from the natural environment to the hive together with floral nectar, pollen, trees’ resins and honeydew [21]. The relationship between the mineral composition of honey and the level of environmental pollution has been confirmed by numerous studies [16].

From an analytical point of view, various methods may have been applied for the determination of the minerals and heavy metals in honey. Among them include inter alia flame atomic absorption spectrometry (FAAS) [22], ionic chromatography (IC) [23,24], pre-separation neutron activation analysis (PNAA) [25,26], flow injection flame atomic absorption spectrometry (FI–FAAS) [27,28,29], inductively coupled plasma optical emission spectrometry (ICP-OES) [17], as well as inductively coupled plasma mass spectrometry (ICP-MS) [30,31,32].

In the current study, the profiling and quantification of potentially toxic elements (PTEs) in honey have been presented. The conducted investigations allowed us to discuss the impact of the botanical origin, geographical traceability and the environmental contamination of the area (soil and plants) from which pollen and nectar for honey production have been gathered. Two sample preparation methods commonly used have also been compared.

## 2. Results

### 2.1. Comparison of Sample Preparation Protocols

Two commonly used preparation methods of honey samples for elemental analysis were compared, namely microwave assisted mineralization and simple dilution. All 45 samples were analyzed. The agreement between content values for 13 PTEs was calculated using a simple formula:(1)$$D=\frac{C_{d}-C_{m}}{C_{d}}\times 100$$
where: D is relative deviation in percent [%]; C_d_ is PTEs content after dissolving in diluted HNO_3_ and C_m_ is PTEs content after mineralization

The mean differences are small ranging from 0.07% to 2.63% for Fe and As, respectively. However, for some honey samples those differences exceed 10% in single analyses of V (2 samples), Cr and Co (1 sample) (Figure 2). Although overall picture clearly indicates that mineralization with HNO_3_:H_2_O_2_ usually gives higher PTEs content, in the majority of samples, those differences were comparable with precisions of measurements expressed as mean relative standard deviation listed in Table 1.

To check the accuracy of ICP-MS measurements and sample preparation protocol, three honey samples of different origin and physiochemical properties were chosen (namely 4, 11 and, 26; *n* = 3) and fortified with 1 mL of 5 ppm, 1 ppm and 10 ppb Mg, Zn and other investigated PTEs, respectively. Spiked samples were diluted to 10 mL with distilled water according to the above-described procedure.

The accuracy was calculated according following formula:(2)$$R\left(\%\right)=\frac{\left(\mathrm{Cx}+s\right)-\mathrm{Cx}}{\mathrm{Cs}}\times 100\%$$
where: Cx is the primary concentration of PTEs in the sample; s is the known amount of PTEs added, Cs is the measured value. Obtained data were presented in Table 1.

We also analyzed certified material Lucerne (RM P-ALFAALFA No. 12-2-03 from Slovak Institute of Metrology, Slovakia) for an additional accuracy check. It has to be mentioned, that we used microwave-assisted digestion protocol for sample preparation described below in Section 4.2. The results are presented in Appendix A.

In view of the above, it was considered that simple dilution in 1% HNO_3_ is sufficient to perform elemental analysis with ICP-MS. It is also worth mentioning that this protocol is much faster, cheaper and does not require the usage of high concentrated acid.

### 2.2. PTEs Content in Honey

The overall content of PTEs in investigated honey samples is presented in Table 2, then in Figure 3, variability plots of PTEs content in honey samples are shown. None of the samples contain cadmium and lead in levels exceeding BEC; therefore, those two elements were excluded from further evaluations. Most of the PTEs (excluding Fe and Zn) have right-skewed distribution, suggesting that the majority of results is below mean value. Arsenic was found in 10 out of 45 samples with highest content (0.49 μg/kg) in polyfloral honey from Turkey and linden honey from Warmian-Masurian Voivodship. Two samples of honeydew honey from Lesser Poland Voivodeship contain elevated content of nickel (>400 μg/kg), which can be related to Ni ore mining activities in that region. In contrast, two other samples from the Warmian-Masurian district contain the highest amount of chromium (3.76 μg/kg) in the case of buckwheat honey and molybdenum (5.94 μg/kg) in the case of dandelion honey.

Cluster analysis (CA) was used to compare the similarities and differences in PTEs composition of honey samples. The tree diagram presented in Figure 4 clearly classifies the samples into four groups. 

First one (marked with yellow) contains honey with the highest content of copper and manganese with mean value of 367 µg/kg (0.367 mg/kg) and 1172 µg/kg (1.172 mg/kg), respectively. On the opposite side of dendrogram, CA creates cluster containing 11 samples of honey with elevated content of arsenic, cobalt, chromium, magnesium, molybdenum, nickel, antimony and vanadium. Honey from some countries other than Poland were grouped in a cluster marked in green (12 out 18 samples). Both clusters located in the middle of the tree graph (indicated by green and blue colors) were characterized by moderate values of analyzed PTEs.

The hypotheses that origin and type of honey is influencing PTEs composition have also been tested for Polish samples (post hoc tests: Tuckey honest significance for unequal N, *p* < 0.05 and Newman Keuls, *p* < 0.05), but no significant differences were found in the case of origin. In contrast to this fact, types of honey are related to different content of zinc (elevated in buckwheat honey and smallest in rapeseed honey), elevated amount of nickel and cobalt in honeydew honey.

### 2.3. Bioelements Content in Honey

Due to the consumption of honey by people around the world, the content of mineral salts, and thus the presence of bioelements in honey, was also important in this study. Based on the information available about the composition of the honey, this natural product contains, in most cases, potassium (up to 500 mg/kg), phosphorus, magnesium and calcium. However, iron, silicon, sulfur, copper, fluoride, zinc and manganese occur in slightly smaller quantities. Moreover, honey contains other important bioelements, such as cobalt, molybdenum, chromium and iodine. The varieties of honey differ, as do the number and content of bioelements, where the average is usually 0.3%, in nectar-type honey it is in the range from 0.01 to 0.035% and in honeydew honey it is approximately 1% [3,5,9]. The content of these metals, which are considered as bioelements, such as magnesium, iron, copper, zinc, manganese, cobalt, molybdenum and chromium, have been highlighted. The results of the comparative analysis are presented in Figure 5.

## 3. Discussion

Investigating several honey samples, we found that simple dilution versus much time-consuming microwave-assisted mineralization is comparable in terms of both precision and accuracy. Comparing the data presented in Figure 5, we noticed that all samples are characterized by a high content of magnesium, particularly goldenrod honey; a sample No. 15, originated from the central part in Poland (Łódź Province); as well as samples of honey No. 6 and 7, which are also polish honeys from Lesser Poland Voivodeship, where concentrations of magnesium have been evaluated 16,080 µg/kg (16.08 mg/kg), 14,980 µg/kg (14.98 mg/kg) and 14,060 µg/kg (14.06 mg/kg), respectively. For a sample of multiflower honey from Turkey (No. 35) there are low concentrations of all determined elements; only the concentration of magnesium at 2860 µg/kg (2.86 mg/kg) was significantly higher than for other elements. The iron content in all cases of test samples was at a similar level and were approximately 1000 µg/kg (1.00 mg/kg). With two exceptions, which were samples of buckwheat honey from the Warmian-Masurian Province in Poland and polyfloral honey from Turkey, these samples display a lack of iron (sample No. 34) or a low concentration 13 µg/kg (sample No. 35) was found.

In addition, the presence of manganese has been considered, and the highest concentration for this bioelement in buckwheat honey sample obtained from the Lower Silesia Province in Poland was evaluated to be 2100 μg/kg (2.10 mg/kg). In addition, the presence of zinc and copper have been studied. The concentrations of these elements were up to several hundred μg/kg, but in the case of some samples (rapeseed honey from central Poland, Kuyavian-Pomeranian Voivodeship, sample No. 16 and multiflower honey from Turkey, sample No. 35), these concentrations were just several μg/kg. However, copper was less frequent than zinc in all tested samples of honey. Exemplary results are presented in Figure 6.

In general, comparing the occurrence of certain elements, which can also be considered as bioelements, concentrations of these in Polish honey were much higher than for foreign honey samples (Figure 6), but a higher number of honeys in Poland have been taken into account in these studies. Furthermore, based on obtained results, the attempt of classification of honey samples according to their provenance, despite their type, has been noticed. The samples of Polish honey are characterized by a high content of nutrients. The natural environment in Poland, characterized by high biodiversity and moderate topsoil contamination compared to other countries in Europe (Figure 6), contributes to a significant diversity of honey varieties. The location of apiaries in industrially uncontaminated areas determines the nutritional and health properties of honey.

The meta-analysis of the source data or results supported by cluster analysis, which are available in the literature, obtained from institutions’ reports related to environmental protection and the results of the research provided by laboratories relating to the presence of potentially toxic elements in honey indicate a group of several of the most frequently occurring elements. Among which the following should be mentioned: iron, manganese, lead, copper, nickel, cadmium, arsenic, and mercury, with iron as the most common element [34]. Iron belongs to the micronutrients necessary for plants’ proper growth and development. The appropriate concentration of iron in the plant is important for the key processes, such as photosynthesis, cell respiration, nucleotide metabolism, or chlorophyll synthesis, which translates into obtaining suitable biomass and nutritional quality for crops in agriculture [35]. Although manganese is a natural component of the soil, and agricultural fertilizers can be considered the most important man-made source of this element. In highly urbanized regions with well-developed agriculture, bees produce honey with a high concentration of chromium, especially due to pesticides and artificial fertilizers [36]. According to the concentration level data, the environmental bioavailability of lead is higher than the concentrations of chromium and cadmium. Relatively low-nickel concentrations as trace elements are necessary for the proper functioning of the human body [37]. Based on studies, honey from Turkey and Brazil contains low concentrations of nickel [29,38]. Hazardous mercury can be present in honey when anthropogenic sources, such as industrial and municipal wastewater, mines, incineration, and the agricultural sector occur near apiaries or hives [39]. When considering honey from Poland, iron, manganese and nickel are usually present but in amounts lower than in other European countries [34]. Usually, the PET sequence is roughly in line with the bioavailability of the PET in the environment. Therefore, the important role of the botanical origin of the honey can be emphasized [34]. However, most studies have shown that plants are significantly exposed to environmental pollution and that bees can transfer metals in the nectar into their hives. Nevertheless, the direct impact of environmental pollution on the quality of honey is often discussed. When determining the ranking order of PTEs, according to their average concentration in honey expressed in a unit [mg/kg], the following summary can be made: Fe > Mn > Pb > Cr > Cu > Ni > Cd > As > Hg [34]. 

## 4. Materials and Methods

### 4.1. Honey Samples Collection

The composition of 45 honey samples in terms of a content of PTEs was compared. Different honey samples used in this study were collected from various parts of Poland (total 33 samples) and other countries, such as Australia (Tasmania State, TAS), Bulgaria, Italy, Germany, Portugal, Romania and Turkey (see Figure 7 and Appendix A). Samples taken into account were received from various flowers, such as rape (4), buckwheat (6), linden (6), acacia (3), goldenrod (4), phacelia (1), dandelion (1). Moreover, there were samples obtained from raspberry flowers (1), sunflowers (2), rosemary (1), bush (1), leatherwood (1), clover (1), as well as polyfloral (10) and honeydew honey (3). Collected samples were placed and stored in glass bottles and kept at room temperature in darkness prior to analysis.

### 4.2. Sample Preparation

The honey samples were prepared in two ways. Microwave-assisted digestion was performed using Nova Wave SA system (SCP Science, Montreal, QC, Canada). An amount of 5 g of honey was dissolved in 5 mL of HNO_3_:H_2_O_2_ mixture (9:1 *v*/*v*) (Suprapure, Merck KGaA, Darmstadt, Germany) and digested. Digestion program and settings are reported in Table 3. Second procedure involves dissolution of 1 g of honey in 10 mL of 1% HNO_3_ (Suprapure, Merck KGaA, Darmstadt, Germany) and that solution was subjected directly to ICP-MS analysis.

### 4.3. Analytical Determination

ICP-MS 2030 system (Shimadzu, Kyoto, Japan) was used to determine 15 trace elements listed in Table 1. Collision mode was chosen to minimize polyatomic interferences. Helium (6 mL/min) and argon (8 L/min) (Air Products, Toruń, Poland) act as collision cell and plasma gases, respectively. Radio frequency power was set at 1.2 kW and collision cell voltage at −21 V. Limit of detection (LOD) and limit of quantification (LOQ) (expressed as 3 × and 10 × of standard deviation) and other validation parameters of the analytical method was evaluated and summarized and presented above Table 1. The precision of each measure (expressed as residual standard deviation RSD) was evaluated in terms of repeatability (*n* = 3) and listed in last two columns of Table 1. The high RSD values were obtained for very low concentrations of PTEs. Calibration curves were obtained by dilution of inorganic quality control standard (IQC-019, Ultra Scientific, North Kingstown, RI, USA) in 1% HNO_3_ (Suprapure, Merck KGaA, Darmstadt, Germany) or in mixture of 1% HNO_3_ (Suprapure, Merck KGaA, Darmstadt, Germany) with 3.5% glucose/fructose, for microwave digestion of direct analysis, respectively. Additionally, the 10 ppb platinum solution as internal standard was constantly supplied by additional tube of peristaltic pump. Both dilution solutions and internal standard were served for correction of matrix effects and signal drift. Correlation coefficients for all calibration curves exceeds 0.998 and LOQs were very low, however background equivalent concentrations (BEC) for cadmium, copper, lead and zinc insisted that working range was higher than few μg/L due to laboratory environment.

### 4.4. Data Analysis

Evaluation and presentation of results were done using Statistica Data Miner 7.0 (Statsoft, Cracov, Poland). Content of PTEs is presented in variability plots. Classification of honey was expressed using cluster analysis with Ward’s method after normalization of obtained data.

## 5. Conclusions

In most cases, the samples used in these studies were characterized by the moderate values of analyzed potentially toxic elements, PTEs. In particular, lead and cadmium were absent in the tested samples in levels exceeding background equivalent concentrations, therefore, those were excluded from further evaluations. The main sources of heavy metals in honey depend on where the bees collect nectar. If nectar is collected from plants growing close to streets with intense car traffic or intensively industrialized areas, consequently, honey from plants located near these areas has a higher content of metals than honey from less industrialized places. As noted, six honey samples from Poland contained the highest concentration of copper and manganese with mean value of 367.00 and 1172.00 μg/kg, respectively. Importantly, four of them were buckwheat honeys. In eleven samples of honey, mainly honeydew honey, we found elevated content of arsenic, cobalt, chromium, magnesium, molybdenum, nickel, antimony and vanadium. Supposing this, these samples could come from areas heavily used by the agriculture and industry. Moreover, proposed sample preparation methods, particularly microwave-assisted mineralization and simple dilution, can be successfully used in honey analysis. Although, simple dilution in 1% HNO_3_ is sufficient to perform elemental analysis with ICP-MS. The obtained results provided the groundwork for further analyses to determine the honey origin and to recognize honey as a biomarker of the environment pollution level in areas where bees collect nectar or honeydew. In spite of that, cluster analysis was used to compare the similarities and differences in PTEs composition of honey samples. Obtained CA results can be used as confirmation that PTEs composition is connected with the origin and type of honey. These investigations provided guidance on the type of honey, as well as on the regions of geographical origin, and should be further discussed, and the monitoring of honey quality should be continued.

## Figures and Tables

**Figure 1 molecules-27-05474-f001:**
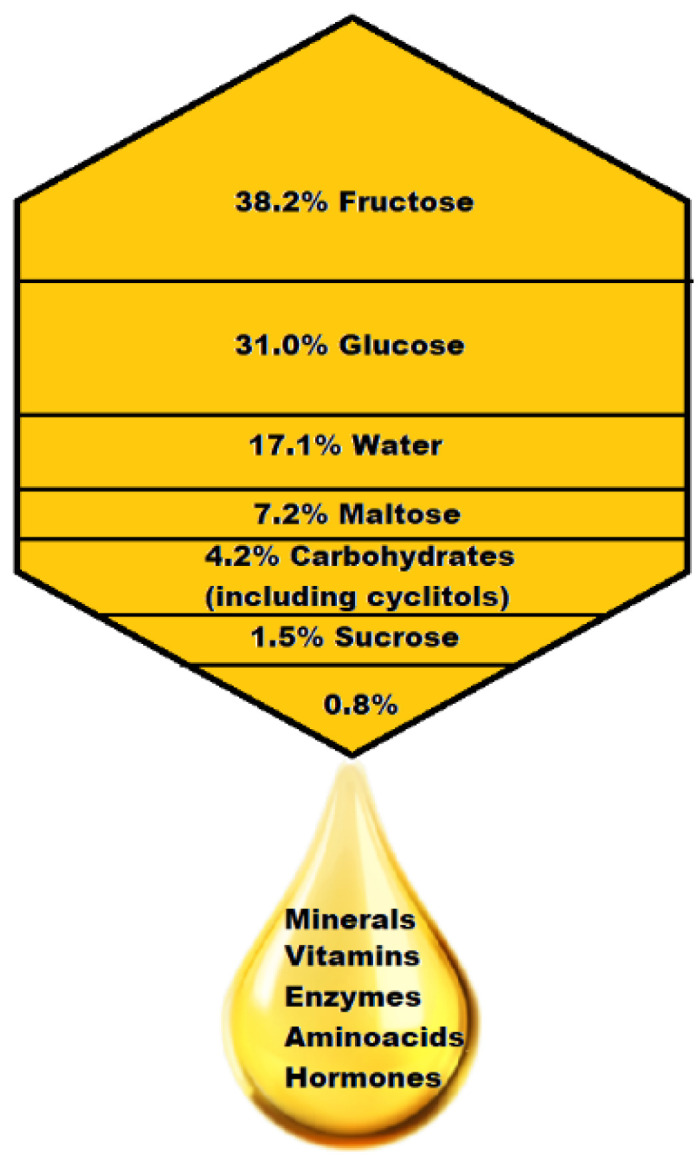
The main groups of compounds found in honey.

**Figure 2 molecules-27-05474-f002:**
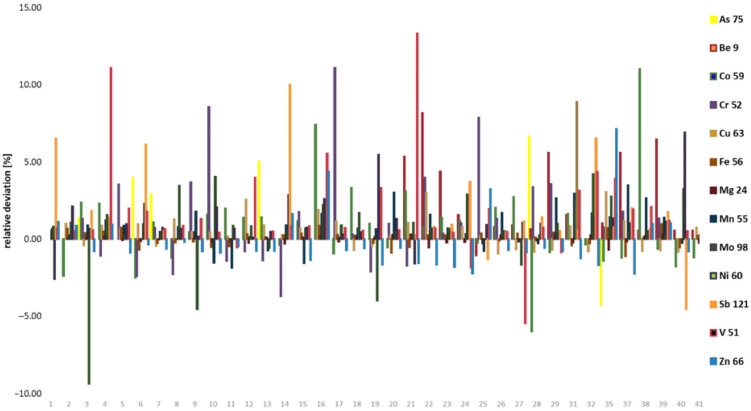
Comparison of sample preparation protocols for all honey samples.

**Figure 3 molecules-27-05474-f003:**
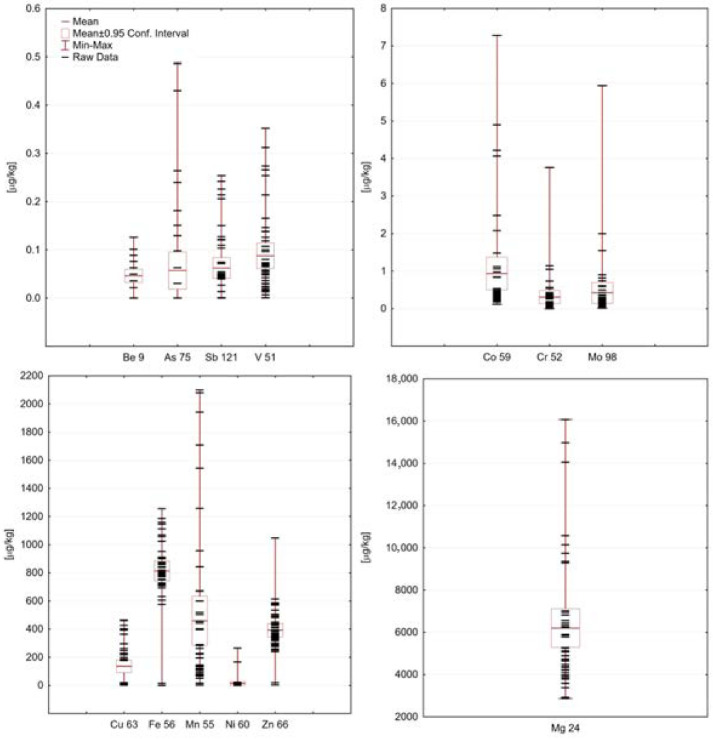
Variability plots of PTEs content [μg/kg] in honey samples.

**Figure 4 molecules-27-05474-f004:**
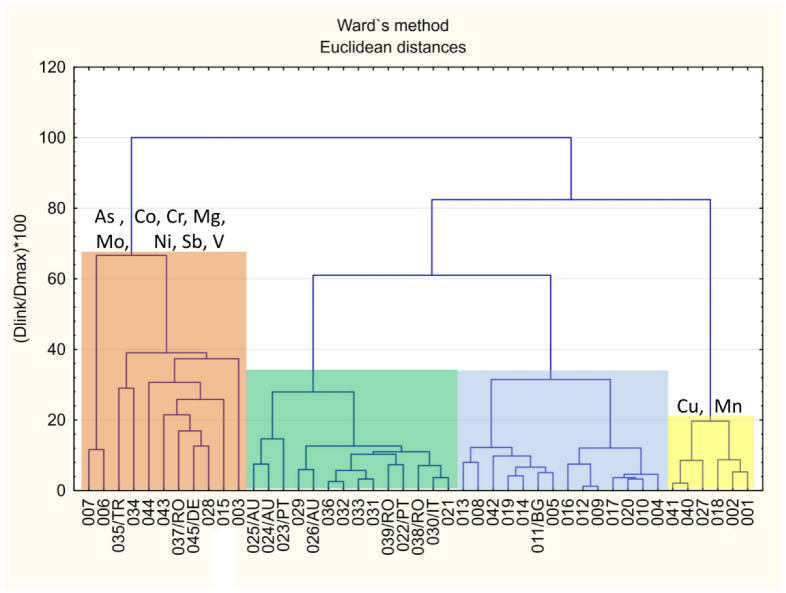
Classification of honey samples according to cluster analysis.

**Figure 5 molecules-27-05474-f005:**
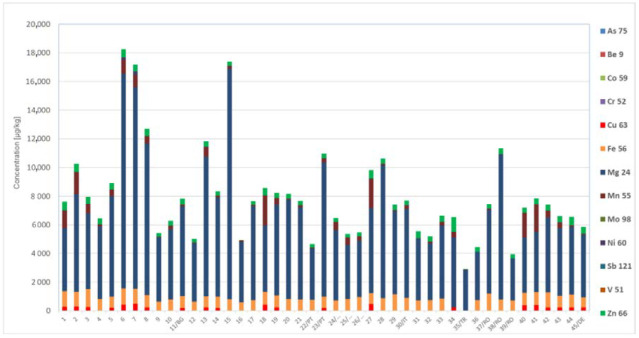
Comparative analysis of honey samples on the content of selected bioelements.

**Figure 6 molecules-27-05474-f006:**
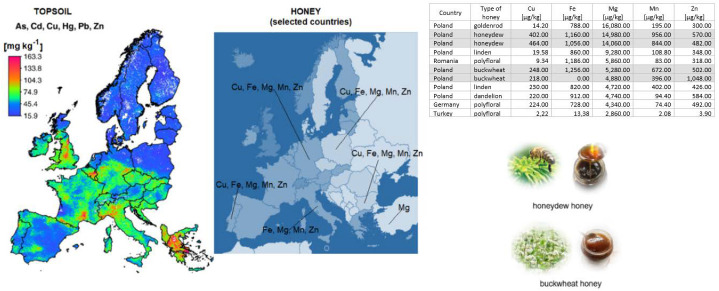
Exemplary results of macro- and microelements concentration in honey samples and maps of heavy metal concentrations in topsoil in Europe, partially according to [33]. Reprinted/adapted with permission from Ref. [33].

**Figure 7 molecules-27-05474-f007:**
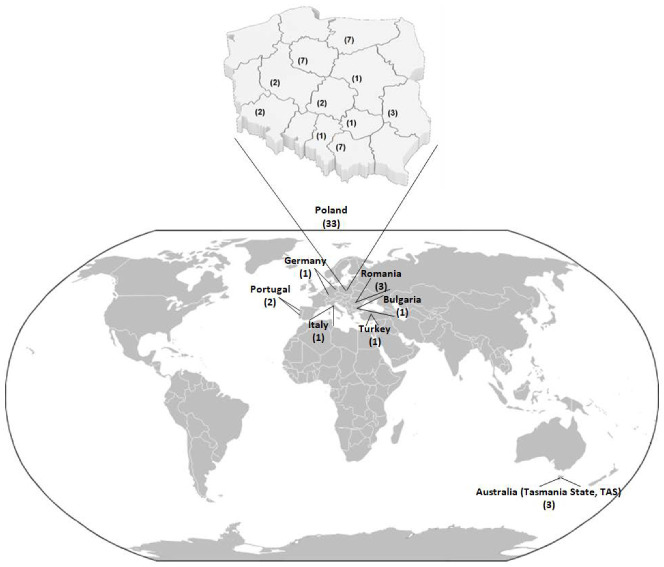
Map of distribution of honey samples in Poland and other countries of the world, including the number of samples.

**Table 1 molecules-27-05474-t001:** Validation parameters of ICP-MS analysis.

Element	Mass	r^2^	Accuracy [%]	LOD [μg/L]	LOQ [μg/L]	BEC [μg/L]	RSD [%] Mean	RSD [%] Max
As	75	1	98.3 ± 2.1	0.008	0.028	0.033	1.9	3.6
Be	9	1	99.9 ± 0.6	<10^−^^3^	<10^−3^	0.042	15.1	36.5
Cd	114	1	97.3 ± 3.1	0.073	0.246	0.101	n.d	n.d
Co	59	0.999	96.8 ± 2.5	0.007	0.025	0.066	7.3	17.2
Cr	52	0.999	98.7 ± 4.3	0.003	0.009	0.128	5.1	17
Cu	63	0.998	100.4 ± 3.9	0.031	0.093	2.013	2.4	17.2
Fe	56	1	100.1 ± 1.1	0.195	0.650	0.745	1.7	9.4
Mg	24	0.998	95.7 ± 6.4	0.075	0.252	0.761	1.6	9.5
Mn	55	1	99.9 ± 4.3	0.064	0.213	0.085	1.6	8.4
Mo	98	1	96.3 ± 3.1	0.004	0.015	0.7	1.9	16.5
Ni	60	1	101.1 ± 2.1	0.012	0.042	0.108	5.6	20.7
Pb	208	0.998	98.2 ± 3.2	0.030	0.101	0.215	n.d.	n.d.
Sb	121	1	101.1 ± 1.8	0.000	0.002	0.057	3.6	10.5
V	51	1	101.2 ± 4.5	0.001	0.003	0.007	2.0	5.4
Zn	66	0.998	97.8 ± 6.1	0.250	0.823	0.341	2.7	18.1

r^2^—determination coefficient; BEC—background equivalent concentration; n.d.—not detected.

**Table 2 molecules-27-05474-t002:** Descriptive statistics of PTEs content [μg/kg] in honey samples.

Element	Mean	Median	Minimum	Maximum
As	0.06	0.00	<LOD	0.49
Be	0.05	0.05	<LOD	0.13
Co	0.93	0.38	0.11	7.28
Cr	0.31	0.14	<LOD	3.76
Cu	136.00	20.00	2.00	464.00
Fe	812.00	798.00	<LOD	1256.00
Mg	6204.00	5120.00	2860.00	16,080.00
Mn	458.00	222.00	2.00	2100.00
Mo	0.42	0.14	0.01	5.94
Ni	13.34	2.18	0.22	266.00
Sb	0.06	0.04	<LOD	0.25
V	0.09	0.06	<LOD	0.35
Zn	393.00	372.00	4.00	1048.00

Element

**Table 3 molecules-27-05474-t003:** Operative conditions for the microwave digestion.

Step	Time [min]	Temperature [°C]	Hold [min]
1	15	120	15 min
2	15	170	10 min
3	20	cooling	-

## Data Availability

Data is contained within the article or Appendix A, additional data is available from the corresponding author upon request.

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
