# Peer review of "Comparative Study of the Potentially Toxic Elements and Essential Microelements in Honey Depending on the Geographic Origin"

_molecules, 2022, doi:10.3390/molecules27175474_

Round 1
Reviewer 1 Report
This work presents a revision of an earlier submission (molecules-1673101). In the revised manuscript all referee comments are properly addressed, and the additional information requested by the reviewers has been implemented. In particular the changes in Table 1 and Table 2 and the more detailed discussion are highly appreciated! Moreover, additional measurements were included to demonstrate applicability of the approach for analysis of reals samples. Summing up, consideration of the referee comments improved the quality but also readability of the work significantly.
Therefore, I recommend to publish the paper in the present form.
Author Response
I am very glad to hear that our manuscript has fulfilled your requirements. On behalf of all authors, we really appreciate all your help.
Reviewer 2 Report
The authors made many of the suggested changes and I think manuscript is much improved. In particular, the Cd and Pb detection limits seem much better after their follow experiments and re-examination of the data. Although again, in absence of any issues, the limits for Cd and Pb should be more similar to that of As; therefore I still believe that there is an underlying issue of some sort, but not sure anything can be done at this point.
For the fortifications, it would be valuable to know what the individual results were – or at least at each fortification level. Ideally the low-level spikes have less accurate recoveries than higher spike levels, so presenting this data would help a reader assess the method’s performance.
For many of the figures, there appear to be two copies/versions in the version that I most recently received. This should be fixed.
Figure 2 –The colors are not very helpful to determine which element is displayed in the graph. Perhaps adding some pattern to those with duplicate colors. Also, the top version does not Sb, V, and Zn.
Table 1 – should be reviewed closely in its final state. I am having a hard time following all the cross-outs, but it appears that Cd and Pb had LOD/LOQ more similar to Mn, Mg, and Zn (elements with traditionally high LOD/LOQ) rather than arsenic and cobalt. Again, I still think this is unusual, but it acceptable. Also, the BEC for Cu seems very high given the relatively LOD/LOQ, please confirm that this is correct. Basically, how can the authors quantify Cu values near their LOQ of 0.093ppb if their background is close to 2ppb.’
Figure 5 – the manuscript mentions that Figure 5 is for bioelements, but the data includes, As, Sb, Be, Ni, and V. The authors description and the figure do not match.
Supplemental Information
Table 1 – the column header (list of elements) should carryover to each page.
Now that all the data has been cleaned up and the results are in the correct units, etc. I was able to read and understand the Results, Discussion, and Conclusion sections more thoroughly. The remainder of my comments have to do with the overall content of the Discussion and Conclusion sections. I am having a hard time understanding the focus/goal of the work. The Discussion and Conclusion sections of a manuscript typically discuss the data, what it means, how it compares to other studies, and what the data will be used for. However, the authors seem to only restate what the results are and offer few reasons/conclusions about why they think the data is how it is. Below are some specific suggestions.
Discussion section
The authors are just reciting the information in SI-Table 1, and I am not sure I understand the point of their discussions. For example – why is it interesting that for #35, only magnesium was high? And what does “higher” mean? Its not high compared to the other samples, is it just higher in concentration than other elements? This type of vague wording should be avoided, and specific criteria should be used. A lot of the discussion seems very general and doesn’t tie into the conclusions very well. Further examples – there were 2 samples with “lower” iron – what does that mean, what is the point of discussing this? The authors do not need to say “manganese” (and other elements) was considered – it was listed in the method and shown in the tables - this seems unnecessary to state that they looked at the data for it – if they didn’t look at it, why would they collect it. The only paragraph that seems to have a clear focus and is useful is the one from lines 208-217.
While the next paragraph 221-250 provides some useful information, the authors do not make any clear connections with their results. During the entire discussion section, there was no discussion/evaluation of the preparation comparison (digestion versus dilution). This was mentioned in the results, but it should be in the discussion. The same for the clustering analysis. The preparation and clustering analysis were one of the few novel parts of this manuscript and the discussion was minimal with little context provided; there was no comparison to other literature reports or how it benefits the field in general, etc. Also, the satisfactory data for the reference material was not discussed.
Conclusion
The authors state some possible conclusions, but it is unclear on what samples they are referencing and their goal is unclear. For example, “Honey from plants located on streets…” but the authors do not state if any of their samples were from streets (which is an odd way to state they are from an industrial/urban area). Again, what does “elevated” mean? The authors state nothing in the conclusion about the accomplishments of the manuscript – one of them being that they show that analyzing honey by MAE and dilution provided similar results. Also, they do not mention the overall findings from their cluster analysis.
Author Response
Reviewer 2
The authors made many of the suggested changes and I think manuscript is much improved. In particular, the Cd and Pb detection limits seem much better after their follow experiments and re-examination of the data. Although again, in absence of any issues, the limits for Cd and Pb should be more similar to that of As; therefore I still believe that there is an underlying issue of some sort, but not sure anything can be done at this point.
We just have annual test of our ICPMS and I (TK) have reported the issue raised by you. Shimadzu does not spot any problems related to As vs Cd after additional tests. I’ll however keep in mind your doubts for next research.
For the fortifications, it would be valuable to know what the individual results were – or at least at each fortification level. Ideally the low-level spikes have less accurate recoveries than higher spike levels, so presenting this data would help a reader assess the method’s performance.
Fortification level has been presented in section 2 lines 103-108. We have tried to fortify our samples with ~3xLOD range.
For many of the figures, there appear to be two copies/versions in the version that I most recently received. This should be fixed.
The reason of this is “track change” version. The upper Figure is erased but still appears in this mode. We suggest to change to No Adjustment in Word’s Reviewer section to see final version.
Figure 2 –The colors are not very helpful to determine which element is displayed in the graph. Perhaps adding some pattern to those with duplicate colors. Also, the top version does not Sb, V, and Zn.
It was also risen by another reviewer. We believe that new version of Fig2 is fulling your requirements. To be true, to present more than 40 samples and 14 elements on one graph is challenging. The idea of it was to prove that simple DI water dilution of sample is still reliable as MAE digestion. I (TK) wish to present, that ~15% is maximum difference between both methods for all elements was investigated.
Table 1 – should be reviewed closely in its final state. I am having a hard time following all the cross-outs, but it appears that Cd and Pb had LOD/LOQ more similar to Mn, Mg, and Zn (elements with traditionally high LOD/LOQ) rather than arsenic and cobalt. Again, I still think this is unusual, but it acceptable. Also, the BEC for Cu seems very high given the relatively LOD/LOQ, please confirm that this is correct. Basically, how can the authors quantify Cu values near their LOQ of 0.093ppb if their background is close to 2ppb.’
In any cases BEC’s were significantly higher than LODs. Iin the case of Cu the highest ratio is related to Cu skimmers used in our instrumentation. Obviously, there is option to use Ni skimmers. Having only one sample (no.35 Table S1) within BEC level while the rest represent much higher values, we decided to work on this ICPMS setup.
Figure 5 – the manuscript mentions that Figure 5 is for bioelements, but the data includes, As, Sb, Be, Ni, and V. The authors description and the figure do not match.
Supplemental Information
Table 1 – the column header (list of elements) should carryover to each page.
This is editorial or technical issue. We’ll, however, put more attention to this table in final version happily published in Molecules.
Now that all the data has been cleaned up and the results are in the correct units, etc. I was able to read and understand the Results, Discussion, and Conclusion sections more thoroughly. The remainder of my comments have to do with the overall content of the Discussion and Conclusion sections. I am having a hard time understanding the focus/goal of the work. The Discussion and Conclusion sections of a manuscript typically discuss the data, what it means, how it compares to other studies, and what the data will be used for. However, the authors seem to only restate what the results are and offer few reasons/conclusions about why they think the data is how it is. Below are some specific suggestions.
We appreciate all your suggestions. We hope that our input fulfils the scope of Molecules and your vision of our work.
Discussion section
The authors are just reciting the information in SI-Table 1, and I am not sure I understand the point of their discussions. For example – why is it interesting that for #35, only magnesium was high? And what does “higher” mean? Its not high compared to the other samples, is it just higher in concentration than other elements? This type of vague wording should be avoided, and specific criteria should be used. A lot of the discussion seems very general and doesn’t tie into the conclusions very well. Further examples – there were 2 samples with “lower” iron – what does that mean, what is the point of discussing this? The authors do not need to say “manganese” (and other elements) was considered – it was listed in the method and shown in the tables - this seems unnecessary to state that they looked at the data for it – if they didn’t look at it, why would they collect it. The only paragraph that seems to have a clear focus and is useful is the one from lines 208-217.
The sentence has been replaced to: “For a sample of multiflower honey from Turkey (No. 35) low concentrations of all determined elements, only the concentration of magnesium at 2 860 µg/kg (2.86 mg/kg) was significantly higher than for other elements”.
While the next paragraph 221-250 provides some useful information, the authors do not make any clear connections with their results. During the entire discussion section, there was no discussion/evaluation of the preparation comparison (digestion versus dilution). This was mentioned in the results, but it should be in the discussion. The same for the clustering analysis. The preparation and clustering analysis were one of the few novel parts of this manuscript and the discussion was minimal with little context provided; there was no comparison to other literature reports or how it benefits the field in general, etc. Also, the satisfactory data for the reference material was not discussed.
Conclusion
The authors state some possible conclusions, but it is unclear on what samples they are referencing and their goal is unclear. For example, “Honey from plants located on streets…” but the authors do not state if any of their samples were from streets (which is an odd way to state they are from an industrial/urban area). Again, what does “elevated” mean? The authors state nothing in the conclusion about the accomplishments of the manuscript – one of them being that they show that analyzing honey by MAE and dilution provided similar results. Also, they do not mention the overall findings from their cluster analysis.
We did substantial changes to those sections. Please refer to new version of the manuscript.
Reviewer 3 Report
Fig. 1 – use unified format of numbers (decimal dot) - still not corrected "38,2% Fructose"
Fig. 2 – the figure designis still confusing - separation space is not between all samples - e.g. 2 and 3... As and Sb have the same colour... I recommend paying attention to better readability of the image.
Author Response
Fig. 1 – use unified format of numbers (decimal dot) - still not corrected "38,2% Fructose"
Fig 1 is corrected.
Fig. 2 – the figure designis still confusing - separation space is not between all samples - e.g. 2 and 3... As and Sb have the same colour... I recommend paying attention to better readability of the image.
It was also risen by another reviewer. We believe that new version is fulling your requirements.
This manuscript is a resubmission of an earlier submission. The following is a list of the peer review reports and author responses from that submission.
Round 1
Reviewer 1 Report
Comments are in the attached file.

Author Response
Thank you very much for your critical review. It was very useful in the correction of our manuscript. Identification of weak points throughout the text has helped us to increase the value of our paper.
The work lacks clearly defined goals. It is not clear whether the aim was to compare the contents of PTEs in different countries, different type of honey, to test different decompositions or show general warnings of honey contamination. The weakness is also the graphic processing of images (see comments below). The text needs to be checked by an English native speaker. I recommend minor changes – focusing to rewrite abstract and conclusion, English correction and graphical correction.
Response: The aim of the work and the conclusions were defined and elaborated, respectively. We put much effort to improve the figures to fulfill standards of Molecules.
Abstract
Key information is missing, such as the analytical method used or a more detailed justification the
context between the selected countries and aim of the research.
Response: According to this comment, required changes were included into manuscript.
- Introduction
English should be corrected, and some sentences should be rewritten because they don't make sense. E.g., “…the variety of elements in honey samples depends on concentrations and it is directly related to the environment the bees are living…
Response: According to this comment, required corrections were done.
Fig. 1 – use unified format of numbers (decimal dot)
Response: According to this comment, required corrections were done.
- Results
2.1. Comparison of sample preparation protocols
There is no information about number of samples that have been tested. Was it for all 45 samples?
Response: All 45 samples were analyzed. This sentence was added to the text.
Fig. 2 – the Figure is not well-arranged. X-axis should give an information about the numbers of tested samples (and the position on the axis) but there is nothing to separate individual samples.
Response: We add separation space between individual samples.
The colors in the chart are easily interchangeable for different isotopes - 4 shades of blue and two purple.
The number of elements does not fit either - 13 are mentioned in the text, there are only 10 of them in the graph.
Response: We redesigned graph according to your suggestions.
The figure needs to be redesigned.
Response: We hope new design fits your suggestions.
Table 1 – the concentrations are in mg/l – it means in solution. So, it is for decomposed sample. How diluted were the samples? The detection limits are quite high – e.g., Pb208 0.446 ppm. Are the units correct?
Response: It is our mistake. While copying to Word we didn’t spot that ug were changed to mg.
Maybe high LOD are the reason for no detection of Pb and Cd in any sample. From the high BEC values: Wasn't there a problem with using contaminated chemicals for sample preparation?
Response: We used ultrapure water and HNO3 acid for ICPMS measurements.
2.2. PTEs content in honey
Table 2 – the minimum values should not be 0.00 ug/kg but bellow the LOD. Use „< LOD”
Response: 0.00 was replaced by < LOD
Fig. 3 – the labels are very small, the label of axes y has only unit, but not what it refers to.
Response: According to this comment, required corrections were done.
Fig. 4 – graphical information about the country of origin is missing. It should be in different colours of the numbers, or each number can be supplemented by the country abbreviation.
Response: Graphical information was added to Fig. 4 as a country abbreviation.
2.3. Bioelements content in honey
Fig. 5 – The colours in the chart are easily interchangeable for different isotopes - 5 shades of blue and 2 purple.
The labels are very small.
Response: We enlarged labels in this graph.
The desired information is unreadable from the picture.
- Discussion
Fig. 6 – The table is too small to be readable
Response: MDPI Molecules is online publisher. We believe that electronic version of figure provides good quality to read table included. All figures were provided in 600dpi. Obviously, the printed draft version can be unreadable.
- Conclusions
The beginning of the conclusion is very general and belongs rather to the introduction.
“In eleven samples 257 of honey were found elevated content of arsenic, cobalt, chromium, magnesium, molybdenum, nickel, antimony and vanadium.“ Are there any norms for the element content in honey (or other food)?
Response: There are no norms of these elements. We just compared those sample’s concentrations with others and mean values.
The conclusion is not specific, and it is not clear what follows from the work.
According to this comment, required changes were included into manuscript.
Reviewer 2 Report
The authors investigated elements of bee honey samples collected from different regions, and deduced the reason of element concentration variation in the samples.
The introduction section did not describe if there is such a knowledge gap which makes the present work interesting. Also, in the discussion section, the authors did not reach any progress by discussing.
Also, it is strange that the authors did not cite any publications in the discussion section. Such a discussion is not a real discussion.
It is not reliable to make a guess of the origin of the high concentrations of the honey sample with only one honey sample and no analysis of other samples, such as soil, dust, or plant.
Author Response
Thank you very much for your review. We are absolutely convinced that your additional comments significantly improved the scientific value of our paper. All comments and changes suggested by Reviewer 2 have been incorporated into the manuscript. Once again, thank you very much for your help.
The authors investigated elements of bee honey samples collected from different regions, and deduced the reason of element concentration variation in the samples.
The introduction section did not describe if there is such a knowledge gap which makes the present work interesting. Also, in the discussion section, the authors did not reach any progress by discussing.
Also, it is strange that the authors did not cite any publications in the discussion section. Such a discussion is not a real discussion.
Response: According to this comment, required changes were included into manuscript. In the discussion section we referred to the current state of knowledge in this area.
It is not reliable to make a guess of the origin of the high concentrations of the honey sample with only one honey sample and no analysis of other samples, such as soil, dust, or plant.
Response: The honey samples were bought locally on markets and stores. We focused on accurate description of origin of honeys. There was no possibility to analyze the other environmental matrices.
Reviewer 3 Report
The manuscript examines trends of various elements for honey samples of different sources. Overall, the manuscript relatively well written and easy to follow. However, I have major concerns about the data and because of this, I strongly think that more experiments are needed. Additionally, I have not reviewed the paper much beyond these areas, as my decision to reject would not be changed.
My issues are with the quality control of the data. The authors mentioned that two preparation procedures were utilized – microwave assisted mineralization and dilution in acid. Their criteria for evaluating the data were based solely on whether the data agreed among the two results, which only tests precision, not accuracy. The authors did not test any certified reference materials or discuss assessing matrix affects using spiked samples (a common issue with just diluting samples as they did for the honey samples). Because their chemometric data is based on quantitative numbers, much more work must be done to affirm these data are correct. Although the authors may have performed high-quality analysis, given the presented information, I have little confidence in the accuracy of the data
The main concern I have is the cadmium and the lead values. The authors make a specific point to discuss them and their absence is a main conclusion from their paper. The data that gathered was not accurate in my opinion in requires more work. The following provides my explanation of the issues.
For Cd and Pb, the detection limits were higher than that for Zn, which I have never seen for an ICPMS method. The detection limits were >0.2mg/L (200 part per billion), which is at least 1000X higher than most typical detection limits for ICPMS. Therefore, it seams as though there was substantial contamination, and if Pb and Cd are that contaminated, this raises concerns that other data is not accurate either. The authors claim that no Pb or Cd was detected at harmful levels, nor above their BEC limits, which we over 1 mg/L (1 part per million), which is higher than any level in honey I have seen reported in the literature for Cd and/or Pb. Since Cd and Pb are primary PTEs which represent half of the “big four” toxic elements that most consider, they are critical elements and must be analyzed properly for this manuscript to be considered.
As I previously mentioned, I did not review the manuscript much more than what is discussed above, but here are few minor issues that I notice during my cursory review of the rest of the document.
The Be calibration curve data seems incorrect – how are the LOD and LOQs ‘0’?
The authors do not explain why these particular elements were chosen – they do not encompass all PTEs nor elements often used for geographic origin (perhaps U and Th should be added, among others?).
Figure 1 seems to add nothing to the paper.
The authors should provide more information on how samples were obtained – did the authors actually collect these samples from hives in a particular location? Or were they purchased from local markets (if so, how was the location verified). The authors appear to make conclusions that samples from countries other than Poland represent the samples of the entire respective country, which may not be appropriate. Perhaps the authors should focus more on the Polish samples and how they are different than the other samples as I am afraid their conclusions may not be supported by their relatively few samples.
Author Response
Thank you very much for your review. We are absolutely convinced that your additional comments significantly improved the scientific value of our paper. All comments and changes suggested by Reviewer 3 have been incorporated into the manuscript.
The manuscript examines trends of various elements for honey samples of different sources. Overall, the manuscript relatively well written and easy to follow. However, I have major concerns about the data and because of this, I strongly think that more experiments are needed. Additionally, I have not reviewed the paper much beyond these areas, as my decision to reject would not be changed.
My issues are with the quality control of the data. The authors mentioned that two preparation procedures were utilized – microwave assisted mineralization and dilution in acid. Their criteria for evaluating the data were based solely on whether the data agreed among the two results, which only tests precision, not accuracy. The authors did not test any certified reference materials or discuss assessing matrix affects using spiked samples (a common issue with just diluting samples as they did for the honey samples). Because their chemometric data is based on quantitative numbers, much more work must be done to affirm these data are correct. Although the authors may have performed high-quality analysis, given the presented information, I have little confidence in the accuracy of the data
The main concern I have is the cadmium and the lead values. The authors make a specific point to discuss them and their absence is a main conclusion from their paper. The data that gathered was not accurate in my opinion in requires more work. The following provides my explanation of the issues.
Response: Thanks for this is a very valuable comment. We did additional experiments with three selected honey samples spiked (fortified) with known amount of PTEs to check accuracy of measurements. New paragraphs were added to results sections. Please see modified Table 1.
For Cd and Pb, the detection limits were higher than that for Zn, which I have never seen for an ICPMS method. The detection limits were >0.2mg/L (200 part per billion), which is at least 1000X higher than most typical detection limits for ICPMS. Therefore, it seems as though there was substantial contamination, and if Pb and Cd are that contaminated, this raises concerns that other data is not accurate either. The authors claim that no Pb or Cd was detected at harmful levels, nor above their BEC limits, which we over 1 mg/L (1 part per million), which is higher than any level in honey I have seen reported in the literature for Cd and/or Pb. Since Cd and Pb are primary PTEs which represent half of the “big four” toxic elements that most consider, they are critical elements and must be analyzed properly for this manuscript to be considered.
Response: This was our mistake. We didn’t check the table after pasting it from another file. Word changed ug to mg automatically.
As I previously mentioned, I did not review the manuscript much more than what is discussed above, but here are few minor issues that I notice during my cursory review of the rest of the document.
The Be calibration curve data seems incorrect – how are the LOD and LOQs ‘0’?
Response: It comes from ICPMS software adjusted by MS Excel. We approximate it to three decimal numbers below ppb. After checking, this value is below ppt and displayed as 0. I has been changed in Table 1.
The authors do not explain why these particular elements were chosen – they do not encompass all PTEs nor elements often used for geographic origin (perhaps U and Th should be added, among others?).
Response: Honestly, it was chosen because of standards available in our lab. We agree that uranium could be interesting to investigate. Our earlier results dealing with soils (analyses done by our scientific partner) indicate high amount of this element due to fertilization of agricultural land. We’ll take this element in further research.
Figure 1 seems to add nothing to the paper.
Response: We decided to keep this figure as general information on honey composition. No other reviewer notice to delete it.
The authors should provide more information on how samples were obtained – did the authors actually collect these samples from hives in a particular location? Or were they purchased from local markets (if so, how was the location verified). The authors appear to make conclusions that samples from countries other than Poland represent the samples of the entire respective country, which may not be appropriate. Perhaps the authors should focus more on the Polish samples and how they are different than the other samples as I am afraid their conclusions may not be supported by their relatively few samples.
Response: The honey samples were bought locally on markets and stores. In most cases, they were local apiaries, and we received honey directly from beekeepers. We focused on accurate description of origin of honeys.
Reviewer 4 Report
The work of Ligor et al. presents the results obtained from the ICP-MS analysis of selected toxic elements in honey. Unfortunately, the paper seems to be rather a technical report than a scientific work. From the analytical point of view this work is an application example for established collision / rection cell technology in ICP-MS, but there are no developments or further improvements reported. For ICP-MS analysis the authors propose simple 1/10 dilution of the honey samples, derived findings were compared with the results from microwave digested samples – indicating only minor differences (less than 10% on average). However, suitability for the analysis of real samples is not demonstrated, usually demonstration of applicability is performed by the analysis of a certified reference material, or by reference measurements with a different technique, or at least with spike experiments and subsequent determination of the recovery rates – none of these approaches were applied in the present study. If the work is considered as an application example an in-depth discussion of the results and a comparison with literature data is missing.
Summing up, publication is not recommended due to a lack of novelty. Additional comments, which underline this decision:
Table 1: are the provided LODs for the diluted honey samples or for the content in the samples, comparison with results from MW digested samples required
Table 2: comparison and discussion of obtained data with literature is missing – e.g. references 22-32
Sample collection:
- detailed description of sample collection - measures to avoid contamination
- storage in glass bottles - exchange of elements possible
- was there any kind of sample stabilization?
Analytical measurements
- sample introduction system - nebulizer suitable for solutions containing 10% dissolved sugar
- ICP conditions - gas flow rates, sample flow rate, plasma power, ...
Author Response
Thank you very much for your review. We are absolutely convinced that your additiona comments significantly improved the scientific value of our paper. All comments and changes suggested by Reviewer 4 have been incorporated into the manuscript. Once again, thank you very much for your help.
4.
The work of Ligor et al. presents the results obtained from the ICP-MS analysis of selected toxic elements in honey. Unfortunately, the paper seems to be rather a technical report than a scientific work. From the analytical point of view this work is an application example for established collision / rection cell technology in ICP-MS, but there are no developments or further improvements reported. For ICP-MS analysis the authors propose simple 1/10 dilution of the honey samples, derived findings were compared with the results from microwave digested samples – indicating only minor differences (less than 10% on average). However, suitability for the analysis of real samples is not demonstrated, usually demonstration of applicability is performed by the analysis of a certified reference material, or by reference measurements with a different technique, or at least with spike experiments and subsequent determination of the recovery rates – none of these approaches were applied in the present study. If the work is considered as an application example an in-depth discussion of the results and a comparison with literature data is missing.
Response: Thanks for this is a very valuable comment. We did additional experiments with three selected honey samples spiked (fortified) with known amount of PTEs to check accuracy of measurements. New paragraphs were added to results and methodology sections.
Table 1: are the provided LODs for the diluted honey samples or for the content in the samples, comparison with results from MW digested samples required
Response: ICPMS values of LODs were compared to be similar between dilution and digestion. We provided the values, which we used for further statistical analysis.
Table 2: comparison and discussion of obtained data with literature is missing – e.g. references 22-32
Sample collection:
detailed description of sample collection - measures to avoid contamination
storage in glass bottles - exchange of elements possible
was there any kind of sample stabilization?
Response: The honey samples were bought locally on markets and stores. We focused on accurate description of origin of honeys. Samples were stored in original packages and deep frozen. Samples were diluted and stored in plastic vials not more than 1h prior the measurements.
Analytical measurements
sample introduction system - nebulizer suitable for solutions containing 10% dissolved sugar
Response: It was discussed with the Shimadzu service that our nebulizer is suitable to operate at such organic content.
ICP conditions - gas flow rates, sample flow rate, plasma power, ...
Response: This information is given in the few first sentences of subchapter 4.3.
Round 2
Reviewer 3 Report
Overall, I appreciate the authors efforts to improve their work. They added in extra experiments to spike sample – which was aimed at testing the accuracy of the method. However, spiking a sample only tests for analyte recovery and matrix effects (which do go towards accuracy, but do not fully test it); but a certified reference material is needed to fully assess accuracy. Also, see comment about Table 1 regarding how the spike recoveries were presented. At the bare minimum this should be done if the manuscript if resubmitted. I think that analyzing a certified reference material (even if it is not honey) would be beneficial to fully test the methods accuracy.
However, I do think there is a big problem with the Cd and Pb data. As mentioned in my first review, having detection limits for Cd and Pb higher than Zn point to a significant contamination issue. Typically, these are on par with As detection limits, and currently they are 100X higher than As. Because of this, I have concerns about the data accuracy, especially for these two elements. Unfortunately, I am unable to accept a document for publication with this type of possible error. The Pb and Cd analysis should be rectified, and samples reanalyzed before submitting the manuscript again.
Below are some additional comments to help with a future submission.
Line 42 – how are pollutants formed in honey? Maybe they are “accumulated”, but “formed” seems to be incorrect. The authors should include a reference for this point or remove it.
Line 102 – this formula is not accurate; it results in s/s = 1. Therefore, the authors should change the symbols and the definitions to be correct. Something like (Cx+s – Cx) / Cs
Figure 2 – the x axis is not readable.
Table 1 – for accuracy appears to be an average of the 3 fortifications, which is not appropriate, the recoveries should be listed individually. For example, spike recoveries of 60, 100, and 140% would average out to be 100%, but the 60% and 140% would not demonstrate accuracy.